# ACT001 Relieves NMOSD Symptoms by Reducing Astrocyte Damage with an Autoimmune Antibody

**DOI:** 10.3390/molecules28031412

**Published:** 2023-02-02

**Authors:** Hongen Li, Mo Yang, Honglu Song, Mingming Sun, Huanfen Zhou, Junxia Fu, Di Zhou, Wenhao Bai, Biyue Chen, Mengying Lai, Hao Kang, Shihui Wei

**Affiliations:** 1Department of Ophthalmology, The Chinese People’s Liberation Army General Hospital & The Chinese People’s Liberation Army Medical School, Beijing 100853, China; 2Department of Neuro-Ophthalmology, Eye Hospital, China Academy of Chinese Medical Sciences, Beijing 100091, China; 3Department of Ophthalmology, The 980th Hospital of the Chinese PLA Joint Logistics Support Force, Shijiazhuang 050082, China; 4Department of Public Health and Preventive Medicine, Shantou University Medical College, Shantou 515041, China; 5Department of Ophthalmology, Beijing Chaoyang Hospital, Capital Medical University, Beijing 100020, China

**Keywords:** neuromyelitis optica, aquaporin 4, mimotope, ACT001, animal model

## Abstract

Neuromyelitis optica spectrum disorder (NMOSD) is a central nervous system inflammatory demyelinating disease, the pathogenesis of which involves autoantibodies targeting the extracellular epitopes of aquaporin-4 on astrocytes. We neutralized the AQP4-IgG from NMOSD patient sera using synthesized AQP4 extracellular epitope peptides and found that the severe cytotoxicity produced by aquaporin-4 immunoglobin (AQP4-IgG) could be blocked by AQP4 extracellular mimotope peptides of Loop A and Loop C in astrocyte protection and animal models. ACT001, a natural compound derivative, has shown anti-tumor activity in various cancers. In our study, the central nervous system anti-inflammatory effect of ACT001 was investigated. The results demonstrated the superior astrocyte protection activity of ACT001 at 10 µM. Furthermore, ACT001 decreases the behavioral score in the mouse NMOSD model, which was not inferior to Methylprednisolone Sodium Succinate, the first-line therapy of NMOSD in clinical practice. In summary, our study showed that astrocytes are protected by specific peptides, or small molecular drugs, which is a new strategy for the treatment of NMOSD. It is possible for ACT001 to be a promising therapy for NMOSD.

## 1. Introduction

Neuromyelitis optica spectrum disorder (NMOSD) is a unique inflammatory demyelinating disease that mainly affects optic nerves and the spinal cord [1,2,3,4]. Formerly known as neuromyelitis optica, NMOSD was considered a subtype of multiple sclerosis (MS) until 2004 when a specific serum antibody, shown to target extracellular conformational epitopes of the water channel protein aquaporin-4 (AQP4) [5,6,7], was detected. The diagnostic criteria for NMOSD incorporate the presence of AQP4-IgG in the patient’s serum [8]. In the central nervous system, AQP4 is restricted to astrocytes and ependymal cells [9]. There are two major AQP4 isoforms, M1 and M23, and both have the same extracellular residues [6,10]. M23 lacks the initial 22 intracellular N-terminal fragments of M1 and aggregates as heterotetrameric particles [11]. The heterotetrameric structure endows the M23 isoform with a higher affinity for AQP4-IgG [12,13]. Evidence from human NMOSD lesion pathology, as well as ex vivo and in vivo data, have demonstrated the pathogenic mechanism of NMOSD, in which AQP4-IgG binds to AQP4 M23 on perivascular astrocyte endfeet, activates the classical complement cascade, and induces granulocyte and macrophage infiltration, leading to secondary oligodendrocyte damage, demyelination, and neuronal death [10,12,14].

There are three AQP4 extracellular epitopes (Loop A, Loop C, and Loop E) and three intracellular epitopes (N terminal, Loop B, and C terminal) [11,15]. The extracellular epitope-targeting antibodies have high NMOSD specificity; within them, the antibodies against Loop C have the strongest sensitivity toward NMOSD [11,16,17]. Interrupting the binding between AQP4-IgG and the AQP4 extracellular epitopes has been hypothesized to protect astrocytes against the immune attack from AQP4-IgG. In animal models, non-complement active antibodies or synthesized drugs can block AQP4 and reduce the cytotoxicity of AQP4-IgG [14,18,19,20,21]. Therefore, we hypothesized that the antigen specificity of AQP4-IgG plays an important role in NMOSD pathology; neutralizing the specific AQP4-IgG may protect astrocytes in NMOSD.

In this study, we synthesized three epitope peptides that have been proven to capture AQP4-IgG from patient serum [11]. Then, these peptides were mixed with serum from AQP4-IgG-seropositive patients. The AQP4-IgG concentration was significantly reduced after incubation with the mimotope peptides when examined with enzyme-linked immunosorbent assay (ELISA) and CBA. In primary astrocyte culture, both the Loop A and Loop C, but not Loop E mimotope, significantly reduced the complement-related cytotoxicity of NMOSD patient serum. In addition, the Loop A and Loop C mimotopes significantly reduced the binding of IgG from patient serum to spinal cord tissue. Moreover, in a behavior test, the Loop C peptide also reversed the motor function impairment and the spinal cord damage induced by the intrathecal injection of NMOSD patient serum. Our in vivo data first demonstrated that the Loop C extracellular epitope of AQP4 is the target of the immune attack in NMOSD pathology. Due to concerns about the druggability and compliance of peptides, we studied the astrocyte protection effect of a natural compound derivative ACT001, which could cross the blood–brain barrier efficiently and produce anti-tumor responses in the central nervous system [22,23]. ACT001 is a water-soluble compound with a stable half-life in plasma, thus making it an attractive candidate for further investigation as a therapeutic in idiopathic pulmonary fibrosis [24]. We revealed that ACT001 could reduce the cytotoxicity of AQP4-IgG in vivo and in vitro, and the treatment effect of NMOSD was not weaker than the first-line drug. In summary, an astrocyte-protection strategy might be a promising therapy for NMOSD and requires further study.

## 2. Results

### 2.1. AQP4 Extracellular Epitope Peptides Reduced the Detectable AQP4-IgG Concentration in NMOSD Patient Serum

Serum samples from 20 AQP4-IgG-seropositive patients were used in this study. AQP4 extracellular epitope peptides were dissolved at 1 mg/mL. Peptide solutions (Loop A:C:E at 1:1:1, or Loop A, Loop C, or Loop E alone) were then mixed with patient serum at a ratio of 1:10 and incubated at 4 °C for 1 h. BSA was used as a negative control. The AQP4-IgG concentration was measured using the aquaporin-4 (AQP4) autoantibody ELISA Kit (RSR Limited, Cardiff, UK). The mixture of Loop A + C + E mimotope peptides significantly reduced the AQP4-IgG concentration detected by ELISA (Figure 1A). When used alone, all three epitope peptides, especially the Loop A and Loop C peptides, significantly reduced the detectable AQP4-IgG concentration (Figure 1A).

To further confirm that the epitope peptides could neutralize AQP4-IgG, we detected the AQP4-IgG at a 1:100-titration in the 20 serum samples after incubation with different peptide solutions (Loop A + C + E or Loop A, Loop C, and Loop E alone). Consistent with the ELISA results, the AQP4-IgG-positive reactions disappeared in samples that previously had a positive reaction after incubation with epitope peptides (except Loop E alone, which did not completely eliminate the positive reaction) (Figure 1B).

Our data demonstrated that AQP4 extracellular mimotope peptides could be recognized by AQP4-IgG, and that the binding of epitope peptides neutralized AQP4-IgG in patient serum. Therefore, we hypothesized that such mimotope peptides could interrupt the binding of AQP4-IgG to astrocytes, thus protecting the cells against complement-dependent cytotoxicity.

### 2.2. Synthesized Epitope Peptides Reduced the Cytotoxicity of NMOSD Patient Serum on Cultured Astrocytes

Individual samples from seropositive AQP4-IgG patients plus active human complement produced a significantly lower lethal concentration 50 than that from healthy controls (Figure 1C, Table 1). When mixed with AQP4-IgG-seropositive samples, Loop A significantly increased the LC50% and right-shifted the cytotoxicity curve (Figure 1C, Table 1); Loop C also increased the LC50% and right-shifted the cytotoxicity curve (Figure 1C, Table 1)—however, Loop E only slightly increased the LC50%, and the cytotoxicity curve was still tied with the control group (Figure 1C, Table 1). The highest LD50% was observed in samples pre-mixed with all three epitope peptides, which had an LD50% close to the healthy serum group, and the cytotoxicity curve was tied with the healthy serum group (Figure 1C, Table 1).

To confirm that these epitope peptides reduced the cytotoxicity of patient serum towards astrocytes by blocking its binding, we performed live-cell immunofluorescence experiments in primary astrocytes. The AQP4-IgG-positive serum mixture resulted in strong immunofluorescent signals on the surface of astrocytes (Figure 1D,E), whereas no IgG binding could be detected on healthy serum-treated astrocytes. Loop A + C + E-incubated serum only labeled a very limited area of astrocytes (Figure 1D). When Loop A and Loop C were used alone, some small IgG-labeled particles could still be detected (Figure 1D,E), whereas there were still many IgG-positive spots in the Loop E group (Figure 1D,E). Our live-cell immunofluorescent results were consistent with the cytotoxicity assay results in this study and further confirmed that the epitope peptides could block the binding of AQP4-IgG to AQP4 in solution and astrocyte primary culture. Our data also implied that the autoantibodies against Loop A and Loop C might be more cytotoxic in NMOSD.

### 2.3. AQP4 Extracellular Epitope Peptides Blocked the Binding of AQP4-IgG to the Spinal Cord Tissue, Thus Protecting the Animal from the NMOSD Model

Before testing on an animal model, we first tested whether the epitope peptides could block the binding of NMOSD patient serum to spinal cord tissue from Lewis rats. After the tissue was incubated with NMOSD patient serum and an anti-GFAP antibody overnight, human IgG antibodies were labeled with a fluorescence-conjugated secondary antibody. The FITC-positive area (GFAP) in the spinal cord also bound the human IgG. This co-localization demonstrated that IgG from NMOSD patient serum bound specifically to astrocytes. When the epitope peptides were added to the serum, the Loop A epitope peptide significantly reduced the human IgG-labeled area at the superficial lamina of the spinal cord. The Loop C epitope peptide almost eliminated the human IgG-labeled area in both white and gray matter. The Loop E peptide only reduced the human IgG-labeled area slightly (Figure 2).

Next, we injected patient serum and human complement into the spinal cord at the L4 level (Day 0). On day 1, the animals began to present with a progressively increasing motor dysfunction score, which peaked on days 3~4 (Figure 3A,B). This result was consistent with previous reports that NMOSD serum plus complement injected into the spinal cord could induce motor dysfunction. Then, the AQP4 extracellular epitope peptides were separately added to the serum specimen mixture 1 h before injection. As our data suggested, incubation with the Loop C peptide resulted in a significant improvement in motor function from the beginning of day 2 (Figure 3A). Furthermore, Loop C peptide incubation drove the motor dysfunction scores back to baseline (levels observed in the healthy serum group) from the beginning of day 4 until the end of the observation period. A *t*-test indicated a large significant difference (*p* < 0.0001) between the Loop C + NMOSD serum group and the NMOSD serum group (Figure 3A). The Loop A peptide also reduced the changes in behavior when compared with the behavior of the NMOSD serum group (Figure 3A). However, the motor behavior of rats treated with the Loop A peptide was still significantly worse than that of the healthy serum group (Figure 3A). The Loop E peptide did not affect the motor dysfunction induced by the NMOSD serum plus complement injection (Figure 3A). Our behavior observations in the NMOSD animal model showed that AQP4 extracellular epitope homologous peptides could protect the spinal cord against NMOSD serum-induced damage. However, post-treatment (injection of mimotopes 24 h after serum) they showed no effect on the NMOSD model (Figure 3B).

We then checked for NMOSD lesions induced in vivo by intrathecal injection of NMOSD serum and human complement into the spinal cord. There was a marked loss of AQP4 and GFAP expression on day 4 after the NMOSD serum injection (Figure 3C), a characteristic feature of human NMOSD lesions and lesions in the mouse spinal cord injected with the NMOSD serum and complement. The loss of GFAP and AQP4 was likely caused by astrocyte damage and death, leading to the loss of myelin basic protein (MBP) (Figure 3C). No loss of GFAP or AQP4 was found in rats injected with the healthy serum control (non-NMOSD) and complement (Figure 3C). The injection of NMOSD serum and the Loop C peptide plus complement together greatly reduced GFAP and AQP4 loss, indicating that the AQP4-IgG plus complement-induced cytotoxicity was blocked in this model. Moreover, the loss of MBP was also reduced when the NMOSD serum was mixed with the Loop C peptide (Figure 3C). These results were consistent with our behavioral observation in the in vivo model of NMOSD. Our findings suggest that AQP4-IgG-induced spinal cord damage could be blocked by Loop C homologous peptides.

### 2.4. ACT001 Could Inhibit the Cytotoxicity of AQP4-IgG

The pathogenic interaction between the astrocyte and AQP4-IgG was the first step of NMOSD. To further explore whether ACT001 could reduce the astrocyte injury induced by AQP4-IgG, cytotoxicity assays were performed.

Our results (Figure 4B) revealed that ACT001 exhibits a concentration-dependent protection effect on the cytotoxicity of AQP4-IgG. We tested the protection activity under ACT001 treatment with different concentrations (0 μM, 2.5 μM, 5 μM, 10 μM, and 20 μM) and found that ACT001 at 10 μM significantly reduced cytotoxicity mediated by AQP4-IgG. The significance level was expressed as follows: * = *p* < 0.05, ** = *p* < 0.01.

### 2.5. ACT001 Alleviates NMOSD Symptoms in the Animal Model

The NMOSD model induced by AQP4-IgG was established to explore the protective effect of ACT001 in vivo. After intragastric injection administration for 15 days, significant behavioral score differences were observed in the ACT001 and MSS groups compared to the PBS group (Figure 4C). On the fifteenth day, the behavioral score was alleviated with 60 mg/kg ACT001 or 30 mg/kg MSS treatment. Finally, the scores of treatments with ACT001, MSS, and PBS were 2.00 ± 0.32, 2.31 ± 0.26, and 3.17 ± 0.26, respectively. In summary, ACT001 significantly reduced the central nervous system inflammation caused by AQP4-IgG in the NMOSD model, and the therapeutic effect was not weaker than MSS.

## 3. Discussion

AQP4-IgG has been demonstrated to bind to the surface of astrocytes and initiate a series of inflammatory cascades that contribute to NMOSD neuropathology [10,13,25]. Different theoretic models have been used to explain how IgG binding induces NMOSD pathology. Some in vitro studies have demonstrated that the binding of AQP4-IgG to astrocytes or AQP4-expressing HEK293 cells induces the internalization of the AQP4–Excitatory Amino Acid Transporter 2 (EAAT2) heterodimer, thereby impairing the water flux and glutamate uptake and leading to excitotoxicity [5,26]. Another model induced an NMOSD-like lesion by injecting AQP4-IgG plus complement directly into the CNS. In this model, internalization of AQP4 was not observed, but the role of downstream inflammatory reactions in NMOSD was studied well [12,14,20,25,27,28]. Both models provided direct evidence that complement-dependent cytotoxicity plays a central role in NMOSD pathology [14,26,29,30]. During in vitro studies, the internalization of AQP4 and EAAT2 was always followed by complement activation [26]. The C1q neutralizing antibody was reported to alleviate NMOSD pathology in animal models by interrupting the complement cascade [20]. Moreover, in the clinic, a pilot study (NCT00904826) indicated that eculizumab, which inhibits complement C5, provided a significant benefit to NMOSD patients by reducing the recurrence rate [31]. In addition to inhibiting complement directly, using non-specific human IgG to induce off-target complement activation can also reduce NMOSD pathology in an animal model. Some synthesized compounds and non-complement toxic antibodies that compete for the AQP4-IgG binding site have been proven to be useful in reducing cytotoxicity and, thus, protect astrocytes in cell culture and animal models [15,21,27,32].

Our results demonstrated that AQP4 extracellular mimotope peptides could inhibit the binding of IgG from NMOSD patient serum to denatured AQP4 fragments and AQP4 on the cell membrane, in fixed tissues, and on living astrocytes. AQP4 extracellular Loop A and Loop C mimotope peptides also reduced the AQP4-IgG-mediated complement cytotoxicity in astrocyte culture. Moreover, pretreatment with Loop A and Loop C mimotope peptides also inhibited the NMOSD-induced motor dysfunction in animal models. However, mimotope post-treatment did not affect the NMOSD model. Obviously, the short half-life of these peptides in vivo may greatly limit the effect. However, some synthesized peptides have also shown a good affinity for AQP4-IgG [33,34]. Developing antibodies without the inherent defects of polypeptides might open a new avenue for NMOSD therapy.

Analysis of the patient specimens revealed that only IgGs against the extracellular loops of AQP4 were specific to NMOSD [5,11,15,16,26]. Iorio, Fryer, Hinson, Fallier-Becker, Wolburg, Pittock, and Lennon [11] used recombined AQP4 extracellular loops to isolate AQP4-IgG in patient serum and revealed that the Loop C antibody was 100% disease-specific, whereas Loop A and E peptides were not. Additionally, an investigation using engineered mutants of human AQP4 revealed that Loop C contains a major epitope that can only be found on high-order array structures [16,17]. All of these reports implied that the IgG against Loop C is the core IgG in NMOSD pathology. In our study, we observed that all three epitope loop peptides investigated could reduce the AQP4-IgG concentration detected by ELISA and CBA. When tested on living astrocytes, both Loop A and Loop C, but not Loop E, resulted in a significant protective effect. The Loop A and Loop C peptides also reduced the binding of patient serum to living astrocytes. However, in spinal cord slices, only Loop C could significantly reduce the binding of IgG from patient serum to the spinal cord. These results were consistent with the previous reports that emphasized the importance of Loop C in AQP4-mediated autoimmunity. Moreover, we researched ACT001, a natural compound derivative that can relieve NMOSD symptoms in vivo and in vitro. Although the latest result shows that microglia play a pivotal role in NMOSD [35], AQP4-IgG mediating the harm to astrocytes remains the first step in NMOSD occurrence. Therefore, we believe that there is strong potential for ACT001 to treat NMOSD. ACT001 has even shown potential as an alternative to traditional MSS treatments. The above experimental results also provided a design idea for ACT001 clinical trials in the future.

## 4. Materials and Methods

### 4.1. NMOSD Serum Collection

Sera from 80 NMOSD patients were collected for this study. The diagnosis followed the NMOSD diagnostic criteria published by the Mayo Clinic [8,9,36,37] and established by our previous work [38,39]. Samples from AQP4-IgG-seropositive patients were used in this study. For the CBA and ELISA tests on the precipitated peptides, 20 individual samples were used. For the cytotoxicity test and primary astrocyte binding assay, 20 samples were mixed for IgG purification and labeling. For the animal models, 4 samples with the highest AQP4-IgG concentration were mixed and used together. Sera from healthy individuals were used in the control group. 

### 4.2. Preparation of AQP4 Extracellular Epitope Peptides 

Peptides corresponding to human AQP4 extracellular epitope Loops A [56–71], C [135–159], and E [205–231] (Figure 5) were synthesized by WuXi APP tec. Inc. (Shanghai, China) and were purified by high-performance liquid chromatography (HPLC). The peptide concentration was determined by Bradford Assay.

### 4.3. AQP4 Transfection and Indirect Immunofluorescence Cell-Based Assay

DNA constructs encoding full-length human AQP4 M23 isoforms were generated by PCR amplification using whole-brain cDNA as a template. M23 AQP4 isoform PCR fragments were linked with a GFP tag, ligated into mammalian expression vector pcDNA3.1, and fully sequenced. HEK293 cell cultures were maintained at 37 °C in 5% CO_2_/95% air in Eagle’s minimum essential medium containing 10% fetal bovine serum, 100 units/mL penicillin, and 100 μg/mL streptomycin. Cells were grown on glass coverslips and transfected with DNA in an antibiotic-free medium using Lipofectamine 2000 (Invitrogen. Carlsbad, CA, USA) according to the manufacturer’s protocol. A pcDNA3.1 vector containing only GFP was transfected as a negative control. Stable AQP4-GFP-expressing, or GFP-expressing clones, were selected following enrichment in Geneticin (Invitrogen. Carlsbad, CA, USA) and plated in 96-well plates at a very low density. Before IgG binding, cells were washed 3 times with normal saline, fixed with 4% paraformaldehyde at 4 °C for 15 min, and then washed another 3 times. Fixed cells were stored at 4 °C in normal saline for several weeks.

The binding of patient serum to the M23 AQP4 isoforms was measured by indirect immunofluorescence imaging. Fixed transfected cells were washed and then incubated with 10% normal donkey serum at 4 °C for 1 h. Then, the normal donkey serum was removed, and serum specimens from patients were added to cells at a 1:100 dilution and incubated at 4 °C overnight. Cells were washed three times on the second day and incubated with the rhodamine-labeled donkey anti-human IgG secondary antibody for 2 h in the dark. Finally, the cells were washed three times with normal saline and restored in phosphate-buffered saline (PBS) containing 15% glycerol. The 96-well plate was then sealed and stored at 4 °C in the dark for several days before microscope examination. Serum samples that labeled cells with both rhodamine and GFP signals on the cell membrane were AQP4-IgG-positive (Figure 6).

### 4.4. Enzyme-Linked Immunosorbent Assay (ELISA)

The AQP4-IgG concentration was also measured using the aquaporin-4 (AQP4) autoantibody ELISA Kit (RSR Limited, Cardiff, UK) following the manufacturer’s instructions.

### 4.5. Animals

Adult (250~320 g) female Lewis rats (Chinese PLA General Hospital Animal Center, Beijing, China) were housed in pairs before surgery in a temperature-controlled room with a 12:12 h light/dark cycle. Food and water were available ad libitum. The experiments were carried out according to the Guide for the Care and Use of Laboratory Animals (The Ministry of Science and Technology of China, 2006), and the proposal of this study was approved by the Experimental Animal Ethics Committee of Chinese PLA General Hospital (No. 2015-x11-53). At the end of the experiment, rats were euthanized with an overdose of sodium pentobarbital.

### 4.6. Primary Astrocyte Culture

The cerebral cortices of Lewis rats (60 days old) were aseptically dissected in HBSS (Hank’s Balanced Salt Solution) containing 0.05% trypsin and 0.003% DNase and were kept at 37 °C for 15 min. The tissue was dissociated for 15 min using a Pasteur pipette and centrifuged at 400× *g* for 5 min. Then, the pellet was resuspended in HBSS containing 40 U papain/mL, 0.02% cysteine, and 0.003% DNase and gently mechanically dissociated for 15 min. Cells were collected by another centrifugation (400× *g*, 5 min), resuspended in HBSS containing only DNase (0.003%), and left for 30–40 min, followed by the collection of the supernatant. The supernatant was then centrifuged for 7 min (400× *g*). The cells from the supernatant were resuspended in DMEM/F12 (10% fetal bovine serum (FBS), 15 mM HEPES, 14.3 mM NaHCO3, 1% fungizone, and 0.04% gentamicin), plated in 6- or 24-well plates pre-coated with poly-L-lysine, and cultured at 37 °C in a 95% air/5% CO_2_ incubator. The cells were seeded at 3–5 × 10^5^ cells/cm^2^.

### 4.7. Cytotoxicity Test

IgG from AQP4-IgG-seropositive patients was purified using the Pierce™ Protein A IgG Purification Kit (Pierce Biotechnology, Rockford, IL, USA). IgG concentrations were measured by the absorbance at 280, adjusted to 1 mg/mL, and stored at −20 °C.

Cytotoxicity of AQP4-IgG was tested using Cell Count Kit-8 (CCK-8, Dojindo Molecular Technologies, Rockville, MD, USA), following the manufacturer’s instructions. In brief, cells were incubated in DMEM with 10% FBS or different concentrations of AQP4-IgG (12 ng/mL, 6 ng/mL, 3 ng/mL, 1.5 ng/mL, 0.75 ng/mL, 0.375 ng/mL, 0.1875 ng/mL, or 0.09875 ng/mL) for 24 h. Then, 10 μL CCK-8 solution was added to each well of the plate. The plate was then incubated for 4 h in an incubator. The absorbance at 450 nm was measured using a microplate reader and normalized to the level of a non-treated cell culture.

Primary astrocytes were inoculated into 96-well cell culture plates (1 × 10^4^ cells/well) and cultured for 6 h. Then, the astrocytes were co-cultured with AQP4-IgG + complement and different concentrations of ACT001 (0, 2.5, 5, 10, and 20 µM) at 37 °C for 1 h. DMEM was used as a negative control, and 0.2% Triton X-100 was used as a positive control. After collecting the cell culture supernatant at 1000× *g* for 10 min, lactic dehydrogenase (LDH) detection reagent was mixed with supernatant in the dark for 15 min according to the Cytotoxicity Detection Kit (Roche, Mannheim, Germany) instructions, and LDH release was measured via a microplate reader (TECAN, Grödig, Austria) at 492 nm. The protective effect of ACT001 against the cytotoxicity of AQP4-IgG was determined with the following formula:LDH release (%) = (ODsample − ODNeg.)/(ODPos. − ODNeg.) × 100%

### 4.8. Live-Astrocyte Immunofluorescence

Primary astrocytes were incubated for 20 min in a live-cell blocking buffer (PBS containing 6 mM glucose, 1 mM pyruvate, 1% BSA, and 2% donkey serum) and then for 30 min with NMOSD patient serum or BSA. Cells were then rinsed extensively with PBS and fixed in 4% paraformaldehyde for 15 min. Then, cells were incubated for 30 min with 4 μg/mL rhodamine-labeled donkey anti-human IgG secondary antibody in a blocking buffer. After incubation with secondary antibodies, cells were rinsed extensively in PBS, and coverglasses were mounted with Hard-set Anti-Fade Mounting Medium (Vector Laboratories, Burlingame, CA, USA).

### 4.9. Intrathecal Catheter Implantation

An i.t. catheter was implanted as described previously. Briefly, under sodium pentobarbital (50 mg/kg, i.p.) anesthesia, a polyethylene catheter (PE-10 tubing, Stoelting, Wood Dale, IL, USA) was inserted through a small hole made in the atlantooccipital membrane and threaded 7.5~8.0 cm caudally into the vicinity of the lumbar enlargement. The rostral part was sutured to the muscle to immobilize the catheter. Animals were given 2 mL lactated Ringer’s solution intraperitoneally following surgery. After intrathecal catheter implantation, animals were housed individually and allowed to recover for 7 days before habituation and testing. Only the animals displaying no discernible motor or sensory deficits were used. The function of the i.t. catheter was tested 1 day after implantation using 10 mL of 2% lidocaine. Its location was also verified at the end of each experiment by dissecting the lumbar spinal cord.

### 4.10. Behavior Test

Animals were observed daily for at least one hour while moving in their cages and moving freely on a large table. Body weight was monitored daily. Motor disability was rated on a spinal cord disease score previously reported (Table 2) [27].

### 4.11. Immunofluorescence

Paraffin sections with a thickness of 5 μm were incubated with normal donkey serum at room temperature for 1 h and then immunostained at 4 °C overnight with antibodies against rat AQP4 (1:200, Santa Cruz Biotechnology (Dallas, TX, USA)), GFAP (1:100, Millipore (Burlington, MA, USA)), myelin basic protein (MBP) (1:200, Santa Cruz Biotechnology), or CD45 (1:10, BD Biosciences (San Jose, CA, USA)), followed by the appropriate fluorescent secondary antibody (1:200, Abcam (Cambridge, UK)). Tissue sections were examined with a Leica (Wetzlar, Germany) DM 4000 B microscope. AQP4, GFAP, and MBP immune-negative areas were quantified using ImageJ.

### 4.12. The Therapeutic Effect of ACT001

The mouse NMOSD model was established by an i.t. catheter implanted as described previously. ACT001 (Accendatech, Tianjin, China) was given to the treatment group at a dose of 60 mg/kg in PBS orally every day for 15 days, and 30 mg/kg methylprednisolone sodium succinate (MSS) (Pfizer, New York, NY, USA), and PBS was given as positive and negative control, respectively. There were six models in the treatment group, eight models in the positive control group, and six models in the negative control.

### 4.13. Statistical Analysis

Comparisons between the two groups were performed using an unpaired *t*-test. *p* < 0.05 was considered statistically significant. The values were presented as the mean ± S.E.M.

## 5. Conclusions

In conclusion, AQP4-IgG against the AQP4 Loop C fragment was central in protecting the NMOSD model. Neutralizing these antibodies could block complement reactions, thus protecting the astrocytes and myelin in the NMOSD model induced by intrathecal AQP4-IgG and the complement. Our results suggested that astrocyte-protection strategies have the potential to become an important research direction for new drug discovery for NMOSD. ACT001 could significantly relieve astrocyte damage induced by AQP4-IgG, and the therapeutic effect was not inferior to MSS. ACT001 could be a promising therapy for NMOSD.

## Figures and Tables

**Figure 1 molecules-28-01412-f001:**
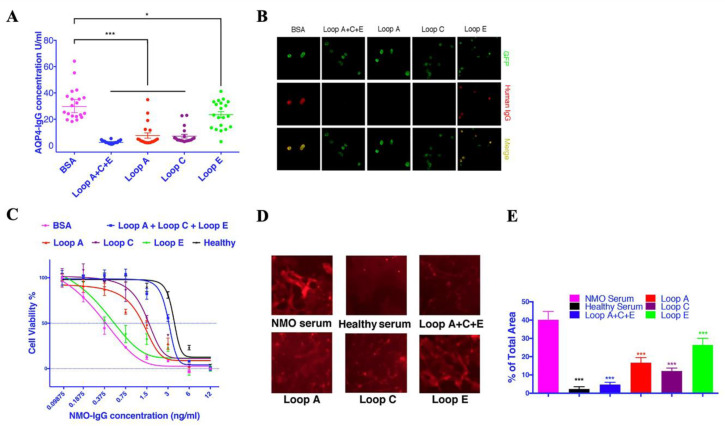
AQP4 extracellular mimotopes neutralized AQP4 autoantibodies. Additionally, different peptides interrupted the binding of AQP4-IgG to cultured astrocytes and reduced the complement-dependent cytotoxicity against astrocytes. (**A**) AQP4 extracellular mimotope peptides reduced the AQP4-IgG concentration detected by ELISA. * represents *p* < 0.05 when compared with the BSA group; *** represents *p* < 0.001 when compared with the BSA group. (**B**) AQP4 mimotope peptides interrupted the binding of AQP4-IgG to AQP4-GFP-transfected HEK293T cells. (**C**) AQP4 extracellular mimotope peptides reduced the complement-dependent cytotoxicity of AQP4-IgG against astrocytes. (**D**) Indirect immunofluorescence revealed that the human IgG-labeled area was reduced when patient serum was incubated with AQP4 extracellular mimotope peptides. (**E**) Relative human IgG immunofluorescence (Mean ± SEM); 4 wells per group. *** represents *p* < 0.001 when compared with the BSA group.

**Figure 2 molecules-28-01412-f002:**
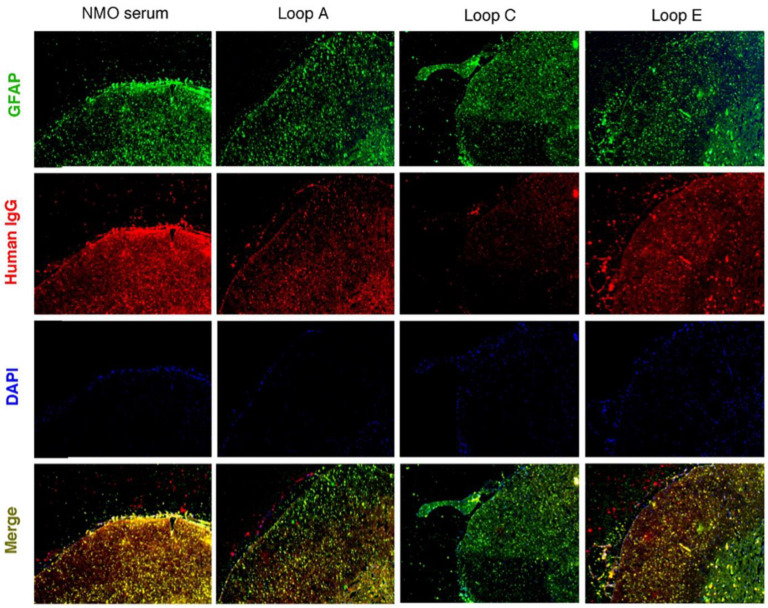
AQP4 extracellular mimotope peptides interrupted serum binding from NMOSD patients to rat spinal cord tissue.

**Figure 3 molecules-28-01412-f003:**
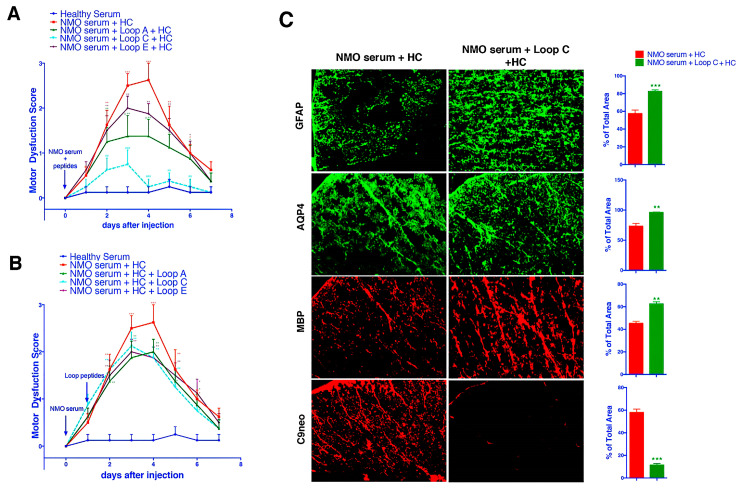
AQP4 extracellular loop mimotope peptides reduced AQP4-IgG-regulated complement cytotoxicity. (**A**) Pretreatment (injected together) with the Loop C mimotope alleviated the motor dysfunction induced by AQP4-IgG plus complement. (**B**) Post-treatment (injected Loop C mimotope 24 h later) failed to reduce the motor dysfunction score. *, **, and *** represent *p* < 0.05, *p* < 0.01, and *p* < 0.001, respectively, when compared with the healthy serum group. #, ##, and ### represent *p* < 0.05, *p* < 0.01, and *p* < 0.001, respectively, when compared with the NMOSD serum group. (**C**) Immunofluorescence staining indicated that the reduction in astrocyte (GFAP and AQP4) and myelin sheath (MBP) was inhibited when AQP4-IgG was precipitated by treatment with the Loop C mimotope peptide, which also reduced the complement reactions.

**Figure 4 molecules-28-01412-f004:**
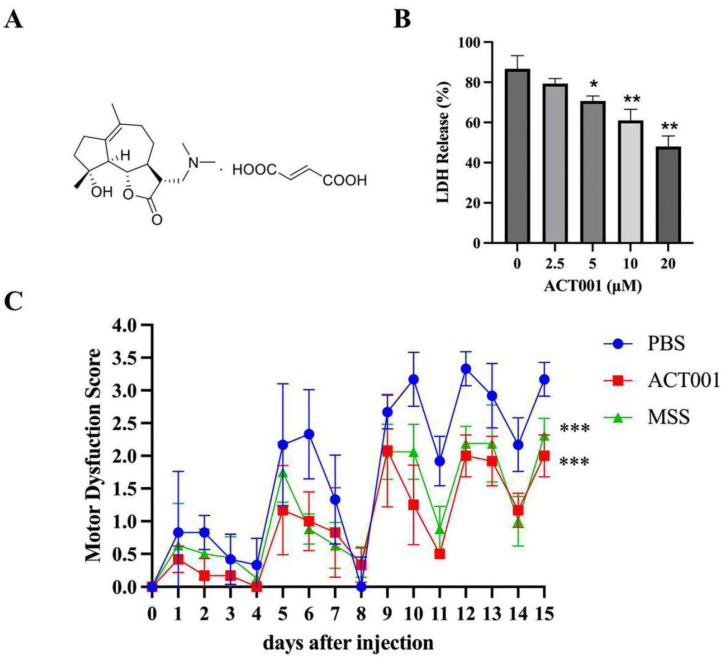
ACT001 reduces the damage caused by autoimmunity antibodies in vivo and in vitro. (**A**) ACT001′s chemical structure formula. (**B**) LDH release in astrocytes by AQP4-IgG with or without ACT001 treatment. * and ** represent *p* < 0.05, and *p* < 0.01, respectively. (**C**) NMOSD model behavior test of the PBS, MSS, and ACT001 treatment groups within 15 days. The significance level was expressed as follows: *** represent *p* < 0.001.

**Figure 5 molecules-28-01412-f005:**
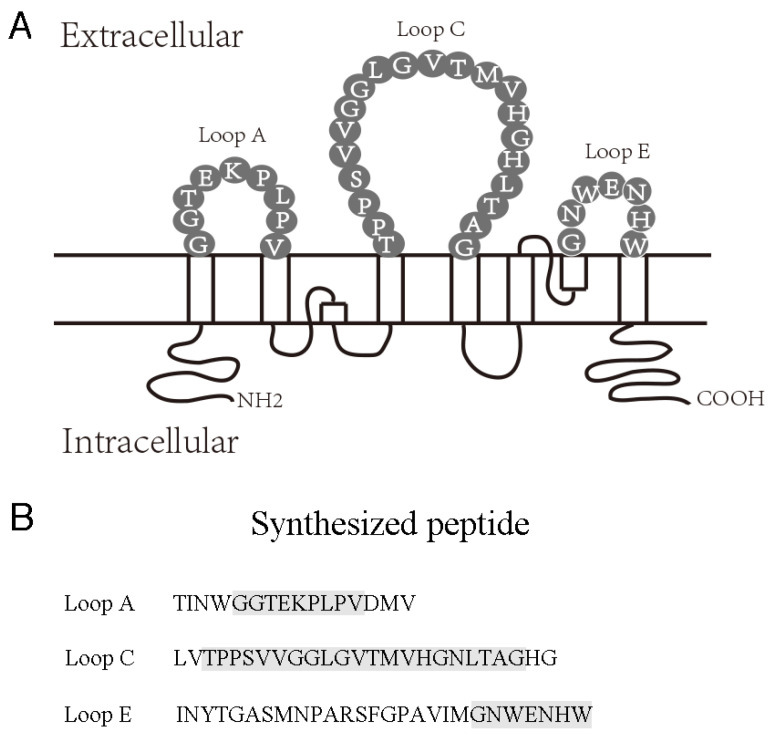
AQP4 extracellular epitopes and synthesized peptides. (**A**) Schematic representation of human AQP4 residues constituting extracellular Loops A, C, and E. (**B**) Synthesized AQP4 extracellular mimotope peptides.

**Figure 6 molecules-28-01412-f006:**
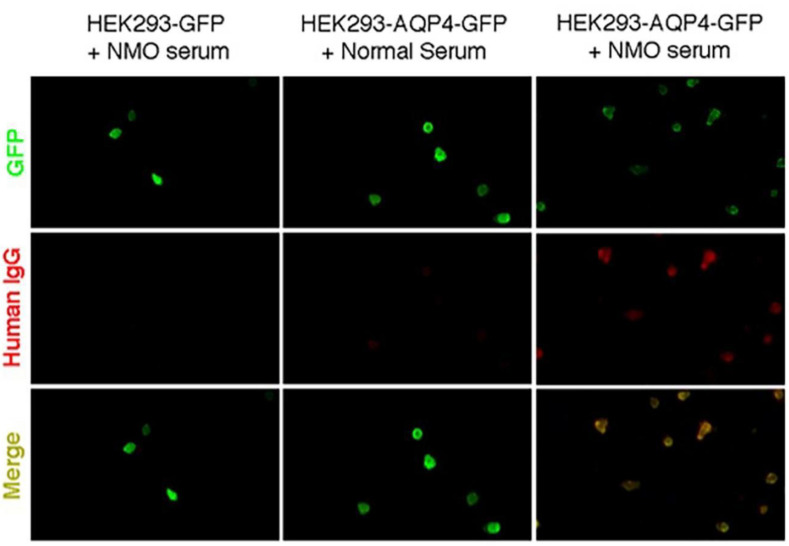
AQP4-GFP-transfected HEK293T cell-based assay of AQP4-IgG in patient serum.

**Table 1 molecules-28-01412-t001:** AQP4 mimotope peptides reduced the complement-dependent cytotoxicity produced by AQP4-IgG from patient serum.

	LC50%	R^2^	*p*
Healthy control	3.602 ng/mL ± 0.1653	0.9555	
AQP4-IgG + Complement	0.396 ng/mL ± 0.056	0.9638	***
AQP4-IgG + Complement + Loop A + C + E	3.052 ng/mL ± 0.1128	0.9512	###
AQP4-IgG + Complement + Loop A	1.253 ng/mL ± 0.1781	0.9019	*,###
AQP4-IgG + Complement + Loop C	1.450 ng/mL ± 0.1879	0.8902	*,###
AQP4-IgG + Complement + Loop E	0.5903 ng/mL ± 0.8902	0.8801	***

* and *** represent *p* < 0.05 and *p* < 0.001, respectively, when compared with the healthy control group. ### represents *p* < 0.001, when compared with the AQP4-IgG + Complement group.

**Table 2 molecules-28-01412-t002:** Experimental neuromyelitis optica-score.

Score Value	Disease Signs
0	Can walk along a bridge without losing balance and can lower itself back onto the ground gracefully using its paws.
1	Loss of footing while walking along the ledge but otherwise good coordination.
2	Ineffective use of hind legs and landing on its head rather than paws when descending onto the ground.
3	Falling while walking or attempting to lower itself or shaking and refusing to move despite encouragement.
4	Both hind legs are paralyzed.

## Data Availability

The datasets used and/or analyzed during the current study can be made available from the corresponding authors upon reasonable request.

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
