# Peer review of "ACT001 Relieves NMOSD Symptoms by Reducing Astrocyte Damage with an Autoimmune Antibody"

_molecules, 2023, doi:10.3390/molecules28031412_

Round 1

Reviewer 1 Report

Response: Wei et al and group explored a study to explain that ACT001 relieves NMO symptoms through reducing astrocyte damage in the presence of autoimmune antibody. Although lot of stupendous work has been done regarding this compound earlier especially in anticancer action as well as anti-inflammatory action. In this paper, investigation of ACT001 as anti-inflammatory in Neuromyelitis optica (NMO) disease reducing astrocyte damage in the presence of autoimmune antibody is novel. Authors have carried out different studies to assess the role of AQP4-IgG in NMO and the role of ACT001 in reducing its cytotoxicity. Thus, the publication is possible after minor revision.

The corrections:

1.     Abstract: The authors have mentioned elaborative introduction related to NMO as well as ACT001 in the abstract. Rather, the abstract should be very concise, clear and should only highlight the outcomes of the research. Other things can be later included in the introduction.

2.     In introduction, justify the importance of the study. Also add whether this mechanistic study is possible for some other inflammatory diseased conditions?

3.     The sequence of the article should include an Abstract, Keywords, Introduction, Materials and Methods, Results, Discussion, and Conclusions as per journal format.

4.     The authors can present a graphical abstract/common figure depicting a synthesized peptide along with their brief result analysis figures from different studies showing anti-inflammatory action.

5.     Connectivity between short sentences can be formed in order to enhance the quality of the paper. For eg- in section 4.2, the lines can be modified as- Peptides corresponding to human AQP4 extracellular epitope Loops A [56-71], C [135-159] and E [205-231] (Fig. 5) were synthesized by WuXi APP tec. Inc. ‘and were’ purified by high-performance liquid chromatography (HPLC).

6.     Authors should consider correct framing of the sentences such as-

a.     in abstract (2nd last line)- ACT001 ‘can protect’ the mouse NMO model instead of ACT001 could protected the mouse NMO model.

b.     Also in acknowledgement, correct sentence should be- All authors ‘have’ read this manuscript and agree with submission.

7.     The conclusion of the manuscript should include the research findings data from the different study assays in order to observe the brief results of the study. Also, mention how these are better in comparison to the reported data.

8.     Reference: Latest references can be included in the manuscript specifically in introduction part such as-

https://doi.org/10.3389/fneur.2020.00501; 10.1016/j.msard.2020.102538 and others.

9.     Manuscript should be checked thoroughly for grammatical and typo errors such as-

a.     in section 2.5 ‘score’ spelling should be corrected instead of sore.

b.     in section 4.12. ‘treatment’ spelling should be checked.

c.     in section 4.13 Values ‘were’ presented as the mean ± S.E.M instead of Values are presented as the mean ± S.E.M.

d.     in Institutional Review Board Statement: All ‘protocols’ involved in this study ‘have’ been approved by Chinese PLA General Hospital instead of All protocol involved in this study had been approved by Chinese PLA General Hospital.

e.     Data Availability Statement: The datasets used and/or analysed during the current study ‘were made’ available from the corresponding authors on reasonable request.

Reviewer 2 Report

The paper presented to me for review however interesting and contributes new knowledge in the relevant area of neurology contains many shortcomings and requires several revisions before it is accepted for publication: 

1. it is not clear from the abstract and from the introduction whether this is a work on humans or on an animal model - both these parts are written abstrusely and should be corrected so that - especially in the abstract - the reader gets basic information about this study

2. the article interchangeably uses the abbreviation NMO and NMOSD - both terms should be clarified, for example, based on the following article, which should be included in the literature: https://pubmed.ncbi.nlm.nih.gov/33802046/

3. it is not clear from the chapter material and method how many people, what genders were qualified for the study? what were the inclusion and exclusion criteria? did the patients have other autoimmune diseases? what was the course of their disease and treatment? did all this affect the results obtained? 

4. there is no sentence about the approval obtained from the bioethics committee  
